# Biogenic Amine Formation in Artisan Galotyri PDO Acid-Curd Cheeses Fermented with Greek Indigenous Starter and Adjunct Lactic Acid Bacteria Strain Combinations: Effects of Cold (4 °C) Ripening and Biotic Factors Compromising Cheese Safety

Charikleia Tsanasidou, Loulouda Bosnea *, Athanasia Kakouri and John Samelis *

Dairy Research Department, Institute of Technology of Agricultural Products, Hellenic Agricultural Organization 'DIMITRA', Katsikas, 45221 Ioannina, Greece; xaroulatsan@gmail.com (C.T.); kakouriathanasia@yahoo.gr (A.K.)
* Correspondence: louloudabosnea@gmail.com (L.B.); jsam@otenet.gr (J.S.)

**Abstract:** The formation of biogenic amines (BAs) in artisan Galotyri PDO cheeses fermented with *Sterptococcus thermophilus* ST1 and the Greek indigenous nisin A-producing *Lactococcus lactis* spp. *cremoris* M78 (A1cheese), or with the A1 starter supplemented with either the enterocin A-B-P-producing *Enterococcus faecium* KE82 (A2cheese) or the multi-functional *Lactiplantibacillus plantarum* H25 (A4cheese) adjunct strains was evaluated. Three pilot-scale cheese trials, GL1, GL2, and GL3, made from boiled ewes' milk, were analyzed for their BA contents before and after cold ripening at 4 °C for 30 days. Total BAs of the fresh GL1 and GL3 cheeses (pH 4.3–4.5) were below 50 mg/kg, except for the A1/GL1 and A1/GL3 cheeses, which contained ca. 300 mg/kg (81.2% histamine) and 1250 mg/kg (45.6% putrescine) BAs, respectively. Whereas due to an outgrowth (>7 log cfu/g) of post-thermal Gram-negative bacteria contaminants during fermentation, most fresh GL2 cheeses (pH 4.7–5.0) accumulated more than 1500 mg/kg of total BAs, which exceeded 3800 mg/kg in all GL2 cold-ripened cheeses due to major increases in cadaverine and putrescine. Tyramine and histamine exceeded 500 mg/kg in the fresh A1/GL2cheeses. Conversely, total BAs remained or declined below 50 mg/kg in all cold-ripened GL3 cheeses. None of the starter or adjunct cultures could be correlated with a specific BA increase, despite *E. faecium* KE82, which increased at 7.6–9.2 log cfu/g in the A2 cheeses is a strong tyramine producer in culture BA broth with 1% tyrosine in vitro. The adoption of strict hygienic measures during artisan Galotyri PDO cheese production (trial GL3) enabled the best performance of all starter LAB strain combinations and reduced BA formation, whereas the high presence of Gram-negative decarboxylating bacteria contaminants compromised cheese (trial GL2) safety.

**Keywords:** Galotyri cheese; biogenic amines; *Lactococcus lactis* subsp. *cremoris* M78; *Enterococcus faecium* KE82; *Lactiplantibacillus plantarum* H25



## 1. Introduction

Biogenic amines (BAs) are non-volatile nitrogenous organic bases of low molecular weight that can accumulate in fermented foods and cause adverse effects on consumer health due to their toxicity [1–3]. Fermented cheeses are among the primary food products most often implicated with BA poisoning because their technology favors BA formation at concentrations high enough to be toxic, depending on the BA type, the cheese type, and consumer sensitivity [1,4,5]. The latter factor is crucial because the ability of the human body to detoxify exogenous BAs ingested with food is highly differential among individuals [1]. Nevertheless, because BA accumulation in foods should be prevented [1,2,5,6], research on this topic has been intensified since 2000 [3]. The relevant knowledge of the technological factors affecting BAs in foods is summarized in recent comprehensive reviews [3], particularly with regard to dairy products [1,4,6–8].

The most important BAs in cheese and other dairy products are histamine, tyramine (i.e., produced by the decarboxylation of histidine and tyrosine, respectively), putrescine (i.e., synthesized via ornithine decarboxylation or agmatine deamination that follows the decarboxylation of arginine to agmatine), and, to a lesser extent, cadaverine, tryptamine, and 2-phenylethylamine (produced by lysine, tryptophan, and phenylalanine decarboxylation, respectively) [3,4,9]. Amongst them, tyramine is the most abundant in cheese [10], causing a migraine crisis known as the '*cheese reaction*'; along with histamine, it causes more severe acute effects on human health [3,11]. However, maximum legal levels have not yet been established for dairy products [6]. Current legislation for BAs in food covers only histamine in fishery products, marketed as either fresh or after enzymatic ripening in brine, with an upper legal limit of 200 and 400 mg/kg, respectively [12]. For all other BAs, the upper levels specified in the EFSA Panel on Biological Hazards (BIOHAZ) report on risk-based control of BA formation in fermented foods are adopted [13].

Linares et al. [4] emphasized that the synthesis of BAs is possible only when three conditions converge: (i) the availability of the substrate amino acids; (ii) the presence of microorganisms with the appropriate catabolic pathway activated; and (iii) environmental conditions favorable to the decarboxylation activity. These conditions in cheese depend on several factors during processing, including temperature, pH, oxygen tension, and NaCl concentration, as well as on process operations affecting the microbial load, such as milk pasteurization or thermization, starter culture type, time and temperature of ripening, and the hygienic practices across processing and preservation [1,3,4,7,10].

Various microbial groups habituated in dairy foods are involved in BA production owing to their enzymatic decarboxylation activities on amino acids derived from casein or other milk proteins [4]. Specifically, among the Gram-negative spoilage bacteria, *Pseudomonas* and *Enterobacteriaceae*, are the major producers of histamine, cadaverine, and putrescine in cheese [1,3,14–18]. *Morganella morganii*, another Gram-negative bacterium, was shown to produce histamine, cadaverine, and putrescine, mainly in the outer layers of French raclette-type cheeses [17,19]. Also, various spoilage yeasts and molds developing on the cheese surface or in the interior of soft cheeses have great potential for putrescine, cadaverine, tryptamine, and histamine production; for instance, *Debaryomyces hansenii*, *Yarrowia lipolytica*, and *Geotrichum candidum* strains [20–22]. BA production is also widespread among Gram-positive bacteria, including coagulase-negative *Staphylococcus* spp. [18], which often are major members of the ripening cheese microbiota [23].

However, the primary BA producers in cheese and other dairy products are lactic acid bacteria (LAB) [10,11]. Fortunately, in most LAB species, the BA-producing ability is strain specific; hence, the selection of starter LAB strains unable to decarboxylate the aforementioned precursor amino acids are feasible and should be promoted toward a safe cheese fermentation and ripening process [1,4]. But still, it remains a safety concern that several nonstarter LAB strains derived from raw milk or various contamination sources before, during, or after cheese processing, such as *Enterococcus*, *Streptococcus*, *Lactococcus*, *Leuconostoc*, and *Lactobacillus*, are capable of in situ production of histamine and putrescine, while, in particular, they are the most efficient producers of tyramine [11]. Very strong tyramine producers are *E. faecalis* and most strains in the *E. faecium/durans* group [24–26]; hence, tyramine production is likely a species-level trait in enterococci [27]. Also, several dairy strains of mesophilic, facultatively heterofermentative nonstarter lactobacilli, particularly *Latilactobacillus curvatus* and *Lacticaseibacillus paracasei*, produce tyramine, histamine, and putrescine [4,11,28]. However, the strongest histamine producer in cheese is the free-living species *Lentilactobacillus parabucnheri* [29–33], while *Levilactobacillus brevis*, another gas-forming LAB commonly found in cheese, produces tyramine and putrescine [34]. Even LAB strain constituents of dairy starter cultures were found to produce BAs [10]. For instance, the ability of certain *Streptococcus thermophilus* and *Lactobacillus helveticus* strains to form histamine [10,35] and of several dairy *Lactococcus lactis* subsp. *lactis* and *Lc. lactis* subsp. *cremoris* strains to form tyramine [36] and putrescine [37] is well documented [4,11]. Additionally to minimizing the above safety risk by excluding BA-producing LAB strains

from well-defined starter cultures, several research studies have indicated another efficient biological approach to reduce BA accumulation: the use of adjunct cultures of selected LAB (*L. casei*, *L. delbrueckii*, and *S. thermophilus*) or yeast (*D. hansenii*) strains capable of degrading the most dangerous BAs, such as tyramine and histamine, in vitro and in situ in cheese and other fermented food products [2,32,38–42].

To date, most traditional Greek cheeses are produced from pasteurized or thermized ewes' or goat's milks, or their mixtures, to eliminate product losses and safety risks associated with the use of raw milk [43,44]. To stabilize cheese fermentation, DirectVatSet (DVS) commercially imported starter cultures (CSCs) comprising either thermophilic, mesophilic, or mixed thermophilic with mesophilic LAB strains are added to the milk post-heating [45,46]. Presumably, all LAB strain constituents in imported CSCs, preselected for their desirable biotechnological properties, should have also been pretested for their inability to produce BAs. Apart from their specific advantages, however, the routine application of the same or similar 'foreign' CSCs in traditional Greek cheese technologies has important disadvantages, such as the growth suppression of the natural raw milk microbiota, major losses of cheese biodiversity, and undesirable alterations in the superior sensorial qualities of artisan Greek cheese products, impairing their authenticity and sustainability [46]. Most negatively influenced are cheese types having a Protected Designation of Origin (PDO) certification and being of major importance for the Greek civil culture, economy, gastronomy, and agro-tourism promotion programs [47,48].

Therefore, to preserve biodiversity, sustainable production, and authenticity of the PDO [44,46,47] and other traditional Greek cheeses not listed in the EU geographical indications registers [49], national supportive actions are (or should be) undertaken. Amongst them, the development of novel 'regional' craft-made cheese starter cultures with improved performance was attempted in the course of the BIOTRUST (2018–2022) research project [50]. For several practical reasons, including the selection of a convenient traditional PDO cheese technology that offers simple and fast production, easy final product packaging and distribution, and rapid income circulation, Galotyri PDO, the oldest soft, Acid-Curd Greek cheese produced exclusively in the regions of Epirus and Thessaly from boiled ewes' or goat's milk and consumed either fresh or ripened [51,52], has served as a real-cheese model to evaluate the in situ performance of newly designed starter and adjunct LAB strain combinations under artisan manufacturing conditions.

Previous Galotyri cheese studies focused on combining indigenous 'regional' antipathogenic strains of nisin-producing (Nis+) *Lc. lactis*, enterocin-producing (Ent+) *Enterococcus*, and non-starter *Lactiplantibacillus plantarum* with potential antifungal and probiotic properties [48,53]. The application of safe, multiple enterocin-producing (m-Ent+) strains of the *E. faecium/durans* group as antilisterial adjuncts in artisan Galotyri PDO production was of particular interest, bearing technological, bioprotective, and potential probiotic benefits; their strong in vitro tyramine-producing ability was the only prominent risk and concern [53,54]. However, whether the most active antilisterial *E. faecium* (also being the strongest tyramine producers in vitro) or the most proteolytic *L. plantarum* ripening strains are able to produce tyramine or other BAs during traditional Greek cheese processing, and storage has yet to be evaluated. Therefore, this study aimed to (i) assess the in situ BA formation in artisan, ready-to-eat (RTE) Galotyri cheeses fermented with selected Greek indigenous starter and adjunct LAB strain combinations; (ii) realize handling operations that may compromise the safety of the fresh cheeses by introducing BA-producing microbial contaminants during the preceding fermentation, drying, and salting steps; and (iii) evaluate the effects of cold ripening on potential changes in the type and concentration of BAs detected in the fresh RTE Galotyri cheeses after packaging and before storage under refrigeration.

## 2. Materials and Methods

### 2.1. LAB Strains, Selection Criteria, and Culture Conditions

Four combinations (A1, A2, A3, A4) of Greek indigenous adjunct and natural starter LAB strains previously monitored for their growth compatibility, bacteriocin gene tran-

scription, and production in model sterile raw milk (SRM) fermentations [55], were tested for their ability to produce BAs in SRM in vitro and then in artisan Galotyri cheese in situ after fermentation and cold ripening. Briefly, A1 was the basic starter LAB combination, consisting of *S. thermophilus* CSL-ST1, a primary acidifying strain of natural origin derived from a freeze-dried CSC product (GRU IDC 01, Centro Sperimentale del Latte [CSL], Lodi, Italy) [56], and the novel, wild nisin A-producing (NisA+) *Lc. lactis* subsp. *cremoris* M78 (GenBank accession number JX402634) isolated from bulk raw ewes'/goats' milk in Epirus, Greece [57]. The A2, A3, and A4 combinations consisted of the A1 supplemented with one 'regional' indigenous adjunct (nonstarter) LAB strain each; A2 with the m-Ent+ (*entA-entB-entP*) *E. faecium* KE82 (MW644969) from a naturally fermented and ripened (3-month-old) Epirus Graviera cheese; A3 with the enterocin A-producing (*entA*) *E. faecium* GL31 (MW709884) from an artisan-type Galotyri PDO retail cheese; and A4 with *L. plantarum* H25, a multi-functional strain from a ripening (5-week-old) Graviera cheese fermented with a mixed thermophilic and mesophilic CSCs [54,55].

Further details on the origin and the main biotechnological and safety attributes of *E. faecium* KE82 and GL31, *L. plantarum* H25, and the two basic starters, *S. thermophilus* ST1 and *Lc. lactis* subsp. *cremoris* M78 strains, are given by Tsanasidou et al. [54]. Relative to the aims of this study, none of them produced histamine after cultivation (37 °C for 72 h) in pure culture in BA broth (pH 5.4) with 1% histidine in vitro [54,58]. However, KE82 was the highest and GL31 was the lowest tyramineproducer among ten autochthonous Ent+ Greek cheese strains of the *E. faecium/durans* group tested under the above culturing conditions in BA broth with 1% tyrosine [54,58]. Because the in vitro formation by the selected starter or adjunct LAB strains of BAs in the presence or absence of precursor amino acids are important preliminary findings for the purposes of this study, the corresponding HPLC-based results of Tsanasidou et al. [54] are recorded in Table 1.

Strains M78, KE82, GL31, and H25 were resuscitated from their frozen (−30 °C) stock culture state in 5 mL portions of de Man Rogosa Sharpe (MRS) broth (Neogen Culture Media; formerly Lab M, Heywood, UK) with 20% (*w/v*) glycerol and activated by two sequent transfers of 100 μLin 10 mL of MRS broth, incubated at 30 °C for 24 h. ST1 was resuscitated and activated twice, as above, in 10 mL of M17 broth (Merck, Darmstadt, Germany) at 37 °C for 24 h. For the preparation of inocula in the SRM or the Galotyri cheese treatments, strains were cultured in appropriate portions of heat-sterilized (121 °C for 5 min) 10% reconstituted skimmed milk (RSM) powder (Neogen, Lab M) and incubated for 24 h at 30 °C or 37 °C, depending on the strain, the milk volume to be inoculated, and the desired inoculation level for each strain in each experiment.

*2.2. Evaluation of the BA Formation Potential of the Selected Starter and Adjunct LAB Strain Combinations in Milk Model Fermentations*

Portions (100mL each) of heat-sterilized (121 °C for 5 min) raw ewe's milk (SRM) in 250 mL Duran flasks, cooled and pre-tempered at 37 °C, were inoculated with each of the aforementioned LAB strain combinations, as described by Asimakoula et al. [55]. Briefly, raw ewes' milk samples (2 liters each) were obtained from our collaborating traditional cheese plant (Pappas Bros., Skarfi E.P.E., Filippiada, Epirus, Greece); heat sterilization was conducted in the microbiology laboratory of the Dairy Research Department (DRD; Ioannina, Greece) to exclude interfering effects of the natural raw milk microbiota on the growth of the A1 to A4 strain combinations in model milk. A fresh 10 mL RSM culture (24 h at 37 °C) of each LAB strain was used to inoculate each SRM flask, as appropriate.

**Table 1.** HPLC-based quantification (mg/L) of biogenic amines (BAs) formed in vitro by pure cultures of Greek indigenous LAB strains in BA broth after culturing at 37 °C for 72 h [1].

| Biogenic Amine | Strain Cultured in BA Broth with 1% Histidine or 1% Tyramine | | | | Strain Culture in BA Broth without Added Amino Acids (Control) | | | |
|---|---|---|---|---|---|---|---|---|
| | *Lc. lactis* ssp. *cremoris* **M78** | *E. faecium* **KE82** | *E. faecium* **GL31** | *L. plantarum* **H25** | *Lc. lactis* ssp. *cremoris* **M78** | *E. faecium* **KE82** | *E. faecium* **GL31** | *L. plantarum* **H25** |
| Cadaverine | 2.05 ± 0.16 | <0.50 | 2.05 ± 0.02 | 2.13 ± 0.18 | 4.10 ± 0.22 | 3.94 ± 0.18 | 7.97 ± 0.13 | 8.16 ± 2.79 |
| Histamine | 16.26 ± 0.88 | <0.50 | 16.00 ± 0.58 | 17.00 ± 1.29 | 13.63 ± 0.00 | 13.29 ± 0.22 | 18.41 ± 0.75 | 18.10 ± 0.40 |
| Putrescine | <0.50 | <0.50 | <0.5 | <0.50 | <0.50 | <0.50 | <0.50 | <0.50 |
| Spermidine | 5.13 ± 0.84 | 1.19 ± 0.00 | 5.88 ± 0.05 | 5.60 ± 0.13 | 10.88 ± 2.65 | 10.16 ± 1.55 | 11.32 ± 0.53 | 21.94 ± 15.83 |
| Tryptamine | 2.83 ± 0.28 | <0.50 | 2.42 ± 0.86 | 2.91 ± 0.22 | 32.26 ± 6.19 | 4.10 ± 0.13 | 10.88 ± 3.27 | 198.38 ± 44.10 |
| Tyramine | 20.22 ± 0.04 | 2706.41 ± 11.51 | 1247.27 ±8.55 | 35.06 ± 21.96 | 31.60 ± 10.39 | 7.28 ± 2.16 | 35.25 ± 22.54 | 187.07 ± 58.78 |
| 2-Phenyl-etlhylamine | 5.58 ± 6.03 | 46.06 ± 0.00 | 21.24 ± 0.77 | 10.35 ± 1.11 | 3.47 ± 0.13 | 76.19 ± 16.79 | 573.32 ± 185.2 | 3.47 ± 0.13 |

[1] All data reported in Table 1 are based on the main datasets or supplementary material published by Tsanasidou et al. [54].

Considering the thermophilic nature of streptococci and enterococci, all flasks were first inoculated with the basic starter A1, at 5.0–5.5 log cfu/mL for ST1 and 6.0–6.5 log cfu/mL for M78, respectively, followed by inoculation of the A2, A3, and A4 flasks with the adjunct strains at 5.0–5.5 log cfu/mL for KE82 and GL31 and 6.0–6.5 log cfu/mL for H25, respectively. All inoculated SRM flasks were incubated at 37 °C for 6 h and then shifted to 22 °C for an additional 66 h to simulate temperature reduction during real cheese milk fermentations. Growth, bacteriocin production, and associated antilisterial activity of each LAB strain in each of the different four SRM cocultures were reported by Asimakoula et al. [55]. For the purposes of this study, BA formation in the clotted (72h) SRM coculture treatments A1 to A4 [55] was determined by the HPLC method previously applied by Tsanasidou et al. [54] for obtaining the BA culture broth data in Table 1. The primary aim was to evaluate whether the two *E. faecium* adjunct strains, KE82 and GL31, could form BAs, particularly tyramine, in model A2 and A3 SRM treatments in comparison to the A1 and A4 treatments, which were free of in vitro tyramine-producing enterococci. The HPLC method for BA determination is described in Section 2.7 below.

### 2.3. Measurement of pH and Determination of Sugars and Organic Acids in SRM Samples

Because BA formation depends on the pH and acidity of the fermented food (dairy) products, the pH of all 72h SRM cocultures (A1 to A4) was measured using a Jenway 3510 digital pH meter (Essex, UK) [55]. Additionally, 2g samples of each SRM coculture were used to determine concentrations (g or mg/100 g) of lactose, D-glucose, D-galactose, L-lactic acid, and acetic acid by the enzymatic kits of R-Biopharm, Boehringer Mannheim (Darmstadt, Germany), according to the manufacturer's instructions for milk products. The pH values and the organic acid concentrations were associated with the levels of BA formation in the 72h SRM cocultures.

### 2.4. Artisan Galotyri PDO Cheese Manufacture and Sampling

Three individual pilot-scale Galotyri cheese production trials, GL1, GL2, and GL3, were processed from boiled ewes' milk (18 kg of raw milk was used for each trial) by following the artisan manufacturing flowchart illustrated by Samelis et al. [48]. The first two trials (GL1 and GL2) were conducted in the Pappas Bros. traditional dairy, while the third trial (GL3) was conducted in the DRD pilot plant to supervise and improve the process hygiene. Each trial included three cheese treatments, A1, A2, and A4, corresponding to the LAB strain combinations, ST1 + M78, ST1 + M78 + KE82 and ST1 + M78 + H25, respectively. The A3 combination (ST1 + M78 + GL31) was excluded from the Galotyri cheese trial experiments of this study because growth, *entA* expression and production, and the associated antilisterial activity of *E. faecium* GL31 were lower compared to those of the m-Ent+ KE82 in A2 or even the A1 and A4 coculture treatments in SRM [55].

Appropriate volumes of fresh (24 h at 37 °C) pure cultures of each LAB strain in RSM were added to the boiled milks (6 kg of milk per batch treatment poured in plastic or stainless steel buckets) to obtain approximate inoculation levels of 6.5–7.0 log cfu/mL for ST1, 6.0–6.5 log cfu/mL for M78, and 5.5–6.0 log cfu/mL for KE82 and H25. In all cheese trials, artisan processing practices and simple equipment were used, which included boiling the milk in an open vessel over a gas burner, cooling it in a basin with running water, and manually pouring the 6kg milk portions into the fermentation buckets [48]. Milk fermentation and the subsequent curd cutting (in 5 × 5 cm cubes), draining (in cheesecloth bags hung at 12 °C for 72 h), dry salting (with 1.5–2.0% edible sea salt) by hand kneading, and packaging (in 5kg plastic containers kept at 4 °C for 48 h to enhance salt diffusion) operations were conducted as described in detail by Samelis et al. [48].

All fresh (day 0) RTE Galotyri cheeses were analyzed for BA contents in correlation with their pH, chemical (gross) composition, and microbiological characteristics. Additionally, all cheeses of the last two trials, GL2 and GL3, were analyzed, as above, after 30 days of storage at 4 °C to assess the potential effects of cold ripening on the BA content and the main cheese quality characteristics tested. In all trials, the A1 treatment was considered

as the control (typical) cheese, whereas the A2 and A4 treatments were considered as new Galotyri PDO cheese specialty products.

### 2.5. Measurement of pH and Gross Composition of the Fresh and Ripened Cheese Samples

The pH of the fresh (day 0) and ripened (day 30) Galotyri cheese products was measured with a Jenway 3510 digital pH meter by immersing the pH electrode directly in small portions of soft cheese mass taken from each plastic container. Appropriate cheese sample portions were also analyzed for moisture, fat, protein, salt, ash, and titratable acidity. Moisture was measured by drying to constant weight at $102\pm1$ °C [59]. Fat content was determined according to the Gerber method [60], protein content by the Kjeldahl method [61], and salt content by the modified Volhard method [62]. Ash content was measured by dry ashing in a furnace at 550 °C [63]. Total titratable acidity was measured by the Dornic method after mixing 10 g of cheese with an equal mass of distilled water and expressed as % ($w/w$) lactic acid [64].

### 2.6. Microbiological Analyses of the Fresh and Ripened Cheese Samples

Portions (25 g) of fresh or ripened RTE cheese samples were aseptically taken from the bulk A1 to A4 cheese containers and homogenized with 225 mL of quarter-strength Ringer solution (Neogen) in stomacher bags (Lab Blender, Seward, London, UK) for 60 s at room temperature. The homogenates were decimally diluted with Ringer, and duplicate 1 mL or 0.1 mL portions of the appropriate dilutions were poured or spread, respectively, on total and selective enumeration agar media. Unless stated otherwise, all media and their supplements were purchased from Neogen. All microbial quantifications were conducted according to the procedures described by Samelis et al. [48].

Total viable counts (TVCs) were enumerated on the Milk Plate Count Agar (MPCA) incubated at 37 °C for 48 h; thermophilic dairy LAB counts (i.e., presumptive streptococci) on the M17 agar incubated at 45 °C for 48 h; mesophilic dairy LAB counts (i.e., presumptive lactococci) on the M17 agar incubated at 22 °C for 72 h; total mesophilic LAB counts on the MRS agar (Neogen, LAB223, pH 5.7; which is highly selective for mesophilic lactobacilli while also supporting the growth of lactococci, leuconostocs, and enterococci) incubated at 30 °C for 72 h; enterococci on the Slanetz and Bartley (SB) agar incubated at 37 °C for 48 h; total staphylococci on the Baird–Parker agar base with egg yolk tellurite (BP) incubated at 37 °C for 48 h; pseudomonad-like bacteria on the *Pseudomonas* agar base supplemented with cetrimide–fucidin–cephaloridine (CFC) incubated at 25 °C for 48 h; coliforms by pouring 1mL samples into the melted (45 °C) violet red bile (VRB) agar, overlaid with 5 mL of the same medium and incubated at 37 °C for 24 h; and yeasts and molds on the rose bengal chloramphenicol (RBC) agar (Merck, Darmstadt, Germany) incubated at 25 °C for 5 days.

The lowest detection limit of all microbial counts was 100 cfu/g of cheese, except for the coliform (VRB) counts that were 10 cfu/g of cheese. The selectivity of the M17, MRS, SB, CFC, BP, and RBC agar media was checked after enumeration by rapid testing of the several representatives of all different colony types for microscopic appearance, and their Gram stain, catalase and oxidase reaction, as described previously [48,65].

To compare the anticipated prevalent colony growth of the inoculated LAB strains as a part of the total LAB growth in all Galotyri cheese samples, selective enumerations were performed. For the A2 treatment, SB/37 °C was the best agar to quantify strain KE82 in the form of typical reddish-brown *Enterococcus* colonies on the SB plates [53]. Whereas accurate selective enumerations of the ST1 and M78 colonies (for all treatments) and H25 colonies (for the A4 cheese treatment) were obtained from the M17/45 °C, M17/22 °C and MRS/30 °C agar plates, respectively, specific colony characteristics of each strain and the growth ability of ST1 at 45 °C were considered, as detailed by Samelis et al. [48]. Additionally, for confirmatory purposes, after enumeration, the high dilution MPCA/37 °C, M17/45 °C, and M17/22 °C agar plates of all cheese treatments were overlayed with fresh cell lawns of *Listeria monocytogenes* No.10 in melted (45 °C) TSAYE [48]. After solidification, the agar plates were incubated at 30 °C overnight. The next day, clear and large *Listeria*

inhibition zones surrounded the NisA+ M78 and m-Ent+ KE82 colonies, if present on the plates, whereas the ST1 and H25 colonies were not inhibitory [48,53].

### 2.7. Determination of Biogenic Amines in SRM and Galotyri Cheese Samples

Determination by HPLC of the BAs produced in the SRM treatments after 72 h and in the fresh (day 0) or cold-ripened (day 30) Galotyri PDO cheese treatments was carried out by acid extraction and derivatization using the method by Eerola et al. [66], as it was applied in pure BA broth cultures of each LAB strain by Tsanasidou et al. [54]. Seven BAs were determined. Standard stock solutions (1 mg/mL) of cadaverine dihydrochloride, histamine dihydrochloride, putrescine dihydrochloride, spermidine trihydrochloride, tryptamine hydrochloride, tyramine hydrochloride, and 2-phenylethylamine hydrochloride, purchased from Sigma-Aldrich Chemie GmbH (Steinheim, Germany) or Acros Organics (Geel, Belgium), were prepared in 50 mL of ultra-pure water (Milli-Q water purification system, Millipore Simplicity UV, France). Working solutions were prepared by diluting appropriate volumes with 10μL of individual standard solution (1,7-diaminoheptane; Sigma-Aldrich) to 1 mL with perchloric acid 70% ($HClO_4$; Fisher Chemical, UK) 0.4 M to obtain concentrations between 0.5 and 5 μg/mL.

For sample preparation, 2 g of each SRM culture or Galotyri cheese together with 125μL of each BA stock solution was homogenized (Ultra-Turrax blender, T25 basic IKA Labor Technik) with 10 mL $HClO_4$, 0.4 M. After centrifugation at $11,300 \times g$ at 4 °C for 30 min, the extraction was repeated with 10 mL $HClO_4$ 0.4 M, followed by centrifugation under the same conditions. The supernatants were adjusted to 25 mL with $HClO_4$ 0.4 M. Derivatization was performed in 1 mL of this sample solution by adding 200 μL NaOH (Lach-Ner, Czech Republic) (conversion of extract to alkaline), 300 μL saturated sodium bicarbonate ($NaHCO_3$) (Fisher BioReagents, Pittsburgh, Pennsylvania, USA) (buffer sample), and 2 mL dansyl chloride (DnsCl) (Sigma). The mixture was transferred to 40 °C for 45 min. The residual DnsCl was removed by adding 100 μL ammonium hydroxide (Fluka, Sigma Aldrich Chemie GmbH, Steinheim, Germany). After 30 min, the mixture was diluted to 5 mL with acetonitrile ($CH_3CN$) (Carlo Ebra Reagents S.A.S, Val de Reuil, France) and centrifuged at $3200 \times g$ at 25 °C for 5 min, and 20 μL of the supernatant was injected into the HPLC.

The BA derivatives (20 μL) filtered through a 0.45μm filter were analyzed on an LC-20AT high-performance liquid chromatographer (Shimadzu, Tokyo, Japan) equipped with a thermostated auto-sampler (SIL-20A), a high-pressure mixing binary pump (LC-20AT), a column oven (CTO-20A), and a diode array detector (SPD-M20A). Separation of the derivatives was carried out on a Shim-pack GIST C18 column (3μm, 100 × 3 mm I.D, Shimazdu, Kyoto, Japan) equipped with a guard column. A gradient elution program with ammonium acetate (Carlo Erba) 0.1M (A) and acetonitrile (B) was used. The gradient started at 50% and ended at 90% acetonitrile in 19 min. The flow rate of the mobile phase was 0.9 mL/min, the column temperature was set at 40 °C, and the peaks were detected at 254 nm. The identification of the BAs was carried out by a comparison of the retention times (RTs) with the aforementioned standard substances. The quantitative determination was carried out according to the method of external standards and integrating peak area in relation to the values of the standard substances. The integration of the area was performed by the HPLC LabSolution software (Version 5.51; Shimadzu Co., Kyoto, Japan). The HPLC system was equilibrated for 10 min before the next analysis [54].

### 2.8. Statistical Analysis

The SRM experiments were replicated twice by analyzing two individual milk samples for each LAB strain combination after 72 h of incubation (*n* = 4). The Galotyri cheese trial experiments were replicated thrice (*n* = 3). For each cheese sample, duplicate agar plates were prepared for counting each microbial group and two measures for each chemical parameter were conducted, and the two values were averaged. The microbiological data were converted to log cfu/g along with the data for each chemical parameter, including the

BAs quantified, which were subjected to a one-way analysis of variance using Statgraphics Plus for Windows v. 5.2 (Manugistics, Inc., Rockville, MD, USA). The means were separated by the LSD procedure at a 95% confidence level (alpha = 0.05) to determine the significance of differences in each cheese treatment with storage time (cold ripening effects), as well as between the cheese treatments on each sampling day (LAB strain combination effects).

## 3. Results

### 3.1. Minimal BA Formation by All LAB Strain Combinations in SRM Cocultures

Minor to undetectable (<0.5 mg/kg) trace amounts of BAs were formed in all model SRM cocultures, A1 to A4, after 72 h of total incubation, including tyramine by *E. faecium* KE82 (traces) and GL31 (8.5 ± 4.3 mg/kg) strains in A2 and A3, respectively. Contrary to our anticipation and the BA broth data (Table 1), the highest tyramine amounts (48.6 ± 23.6 mg/kg only) were found in the A4 cocultures of the basic starter with the *L. plantarum* H25 adjunct strain rather than in the A2 or A3 cocultures of the basic starter with each of the *E. faecium* adjunct strains. Neither KE82 and GL31 nor M78 and H25 strains formed detectable amounts of any of the remaining six BAs tested, including histamine, in SRM fermentations; for this reason, these 'flat' (no BA detection) data are not tabulated. Particularly, the m-Ent+ *E. faecium* KE82 did not form detectable (>0.5 mg/kg) tyramine in the fermented A2 model milk, despite the fact that it was the strongest tyramine-forming (2076.4 mg/L) strain in the BA broth with 1% tyrosine in vitro (Table 1). The formation of BAs, particularly tyramine, by the KE82 and GL31 strains in the A2 and A3 treatments was likely prevented by the rapid and extensive milk acidification, as indicated by the low pH and major L-lactate accumulation in all 72h SRM cocultures (Table 2).

**Table 2.** pH values and concentrations of main sugars (g/100 g) and organic acids (mg/100 g) in model sterile raw milk (SRM) cocultures of starter with adjunct LAB strains after a total incubation (milk fermentation) period at 37 °C to 22°C for 72 h [1].

| Chemical Parameter | SRM Treatment | | | |
|---|---|---|---|---|
| | A1 (ST1 + M78) | A2 (ST1 + M78 + KE82) | A3 (ST1 + M78 + GL31) | A4 (ST1 + M78 + H25) |
| Milk pH [2] | 4.25 ± 0.04 ab | 4.28 ± 0.05 b | 4.29 ± 0.04 b | 4.18 ± 0.11 a |
| Lactose | 2.88 ± 0.13 ab | 3.00 ± 0.12 b | 2.78 ± 0.17 ab | 2.67 ± 0.30 a |
| D-glucose | Traces | 0.02 ± 0.02 a | 0.02 ± 0.01 a | 0.04 ± 0.02 a |
| D-galactose | 0.98 ± 0.11 b | 0.90 ± 0.10 a | 0.84 ± 0.11 a | 0.94 ± 0.09 ab |
| L-lactate | 640.3 ± 167.2 b | 734.2 ± 23.6 b | 747.6 ± 26.3 b | 402.0 ± 147.2 a |
| Acetate | 13.6 ± 3.5 a | 22.8 ± 3.5 b | 33.0 ± 2.2 c | 20.0 ± 0.4 b |

[1] Values are the means ± standard deviation of two individual sample measurements from two replicate experiments (*n* = 4); within a row, means with different letters are significantly different (*p* < 0.05). ST1 + M78: *S. thermophilus* ST1 + *Lc. lactis* subsp. *cremoris* M78; ST1 + M78 + KE82: *S. thermophilus* ST1 + *Lc. lactis* subsp. *cremoris* M78 + *E. faecium* KE82; ST1 + M78 + GL31: *S. thermophilus* ST1 + *Lc. lactis* subsp. *cremoris* M78 + *E. faecium* GL31; ST1 + M78 + H25: *S. thermophilus* ST1 + *Lc. lactis* subsp. *cremoris* M78 + *L. plantarum* H25.[2] The pH data after 72 h of incubation were adapted from Asimakoula et al. [55].

The initial pH of SRM before inoculation was 6.53, and it contained 4.57% lactose, 0.01% glucose, 0.05% galactose, and 24.3 and 3.1 mg/100 g of L-lactate and acetate, respectively. Major decreases in lactose and increases in galactose, L-lactate (mainly), and acetate concentrations occurred after the 72h milk fermentation period (Table 2). Concurrently, the curdled milk pH dropped to 4.2–4.3 due to the predominant growth of the primary acidifying strain, *S. thermophilus* ST1, in all SRM coculture treatments [55]. Of note, the 72h L-lactate content in the A4 treatment was remarkably lower than in the A1, A2, and A3 treatments (Table 2), probably because the *L. plantarum* H25 adjunct strain formed DL lactate racemase; the D-lactate isomer was not determined for the purposes of this study.

*3.2. Variations in pH and the Gross Composition of the Galotyri PDO Cheese Samples Attributed to Certain Unstable Processing Factors of the Artisan Manufacturing Method*

Table 3 summarizes the results of pH and gross composition of the artisan Galotyri PDO cheese samples before (day 0) and after (day 30) cold ripening storage at 4 °C in air (i.e., in plastic 5 kg containers with the soft cheese mass covered with a food-wrapping film membrane to prevent surface mold growth). All fresh RTE cheeses, analyzed two days after dry salting, had a pH <5.0, well above pH 4.0; the lowest and highest pH values were 4.33 (A1/GL3) and 4.97 (A2/GL2), respectively. The mean pH 4.47–4.60 of the fresh cheese treatments did not differ significantly (Table 3). However, a replicate trial effect was evident, irrespective of the LAB strain combination used for cheese making. Specifically, the pH of the fresh A1, A2, and A4 cheeses in the GL2 trial was 4.72, 4.97, and 4.62, respectively, whereas the fresh cheeses in the GL3 trial were more acidified, having pHs of 4.33, 4.34, and 4.35, respectively, and those in the GL1 trial had an intermediate pH value range of 4.44 to 4.52. The above trial-dependent rather than treatment-dependent pH variations indicated that milk acidification was variable. Thus, the artisan Galotyri cheese fermentation was quite an unstable process for reasons that will be addressed in later sections. Conversely, no significant differences in the mean pH of either treatment occurred after 30 days of storage at 4 °C (Table 3), indicating that cold ripening had limited effects on altering the pH of the fresh Acid-Curd cheeses.

**Table 3.** Physicochemical characteristics of artisan Galotyri PDO cheeses fermented with three (A1, A2, and A4) different Greek indigenous starter/adjunct LAB strain combinations [1].

| Parameter | Artisan Galotyri Cheese Treatment | | | | | |
|---|---|---|---|---|---|---|
| | A1 (ST1 + M78) | | A2 (ST1 + M78 + KE82) | | A4 (ST1 + M78 + H25) | |
| | Fresh Cheese (Day 0) | Ripened Cheese (Day 30) | Fresh Cheese (Day 0) | Ripened Cheese (Day 30) | Fresh Cheese (Day 0) | Ripened Cheese (Day 30) |
| pH | 4.52 ± 0.20 a | 4.50 ± 0.17 a | 4.60 ± 0.33 a | 4.50 ± 0.24 a | 4.47 ± 0.14 a | 4.47 ± 0.19 a |
| Moisture (%) | 67.5 ± 6.6 a | 63.5 ± 7.4 a | 68.3 ± 4.5 a | 64.7 ± 7.2 a | 68.3 ± 4.8 a | 65.9 ± 4.4 a |
| Fat (%) | 13.9 ± 2.5 b | 13.8 ± 1.8 b | 13.5 ± 1.8 b | 13.2 ± 1.6 ab | 13.0 ± 1.4 ab | 12.4 ± 0.6 a |
| Protein (%) | 12.9 ± 0.9 c | 11.8 ± 0.4 bc | 10.8 ± 0.3 a | 11.4 ± 0.4 ab | 10.9 ± 0.9 a | 10.6 ± 1.5 a |
| Salt (NaCl) (%) | 1.73 ± 1.13 a | 2.31 ± 1.30 a | 1.56 ± 0.64 a | 2.07 ± 0.64 a | 1.58 ± 0.69 a | 1.87 ± 0.68 a |
| Ash (%) | 2.50 ± 1.11 a | 2.83 ± 1.09 a | 2.26 ± 0.65 a | 2.57 ± 0.43 a | 2.21 ± 0.71 a | 2.50 ± 0.64 a |
| Acidity (%) | 1.07 ± 0.20 a | 1.04 ± 0.03 a | 0.99 ± 0.20 a | 1.14 ± 0.31 a | 0.96 ± 0.07 a | 1.06 ± 0.13 a |

[1] Values are the means of three independent cheese trials (*n* = 3); within a row, means with different letters are significantly different (*p* < 0.05). ST1 +M78: *S. thermophilus* ST1+*Lc. lactis* subsp. *cremoris* M78; ST1 + M78 + KE82: *S. thermophilus* ST1 + *Lc. lactis* subsp. *cremoris* M78 + *E. faecium* KE82; ST1 + M78 + H25: *S. thermophilus* ST1 + *Lc. lactis* subsp. *cremoris* M78 + *L. plantarum* H25.

Similar to pH, the mean titratable acidity values of the cheeses, which ranged from 0.96% to 1.07%, did not differ significantly between treatments before (day 0) or after (day 30) cold ripening. Also, no major differences were found in the mean (%) moisture, fat, protein, salt, and ash content between the fresh and ripened cheeses of each treatment or between the A1, A2, and A4 treatments on each sampling day (Table 3). However, certain constant trends were noted; for instance, (i) compared to the fresh cheeses, the moisture (%) of the ripened cheeses decreased by ca. 2.5–4.0%, whereas salt increased by ca. 0.3–0.6% and ash increased by ca. 0.3% in all treatments; and (ii) while the effects of cold ripening on fat and protein within each treatment were minor, the A1 cheeses contained ca. 1.0–1.5% more fat and 1.0–2.5% more protein (*p* < 0.05) than the A4 cheeses. Nevertheless, none of the above changes was major because (i) all cheeses were stored at 4 °C in water-impermeable plastic containers, and thus, further draining (moisture loss) from the samples during cold ripening was limited; and (ii) regardless of treatment, the cheese trial replicate effects on moisture, salt, and ash contents were even stronger than those reported above for pH (viz., the high SD values in Table 3). So, the fresh GL2 cheese samples had the lowest moisture and the highest salt contents, which were 60, 63.1, and 63.0%, and 2.99, 2.27, and 2.32%, for A1, A2, and A4 treatments, respectively. Conversely, the fresh GL1 cheese samples had the highest moisture (72.6, 71.3, and 72.4%) and the lowest salt (0.81, 1.04, and 0.95%),

respectively, while the fresh GL3 cheese samples had intermediate moisture (69.6–70.5%) and salt (1.38–1.48%) contents before storage.

To sum up, all fresh cheeses in the GL2 trial were much more dehydrated than the respective samples in the other two trials, despite the fact that they simultaneously had the highest pH values. Altogether, the above findings indicated that the drying and salting operations of the acidified curds during the artisan processing of the first (GL1) and mainly the second (GL2) Galotyri cheese trials in the commercial plant were also unstable and caused even greater RTE product variations than the preceding cheese milk fermentation process. One of the primary reasons that the third trial (GL3) was conducted in the DRD pilot plant was to improve control of the fermentation, drying, and salting operations.

### 3.3. Microbial Quantification in the Artisan Galotyri PDO Cheese Samples

The microbial quantification data on the basis of the total and selective enumeration procedures applied are presented in Figures 1 and 2 and Table 4. Specifically, to facilitate data comparisons, Figure 1 shows the total viable counts (TVCs; on MPCA/37 °C) and the populations of total mesophilic LAB (on MRS/30 °C agar), enterococci (on SB/37 °C agar), and yeasts (on RBC/25 °C agar) in the fresh (day 0) and ripened (day 30) A1, A2, and A4 cheese treatments, respectively. Figure 2 shows the populations of thermophilic or mesophilic dairy LAB on M17/45 °C and M/17/22 °C agars, respectively, along with the corresponding basic starter ST1 and M78 strain populations for the A1, A2, and A4 cheese treatments, selectively enumerated on the above M17 agar plates. The populations of the two adjunct strains, KE82 and H25, which were major parts of the respective total mesophilic LAB and *Enterococcus* populations in the A2 and A4 treatments only (Figure 1), are specified in the text. The populations of *Pseudomonas*-like bacteria (on CFC/25 °C agar) and coliform bacteria (on VRB/37 °C agar) in the A1, A2, and A4 treatments are tabulated separately for each Galotyri cheese trial in Table 4 for reasons addressed below. The quantification data for total staphylococci are not plotted or tabulated because their populations were below 3 log cfu/g in all cheese treatment and trial samples.

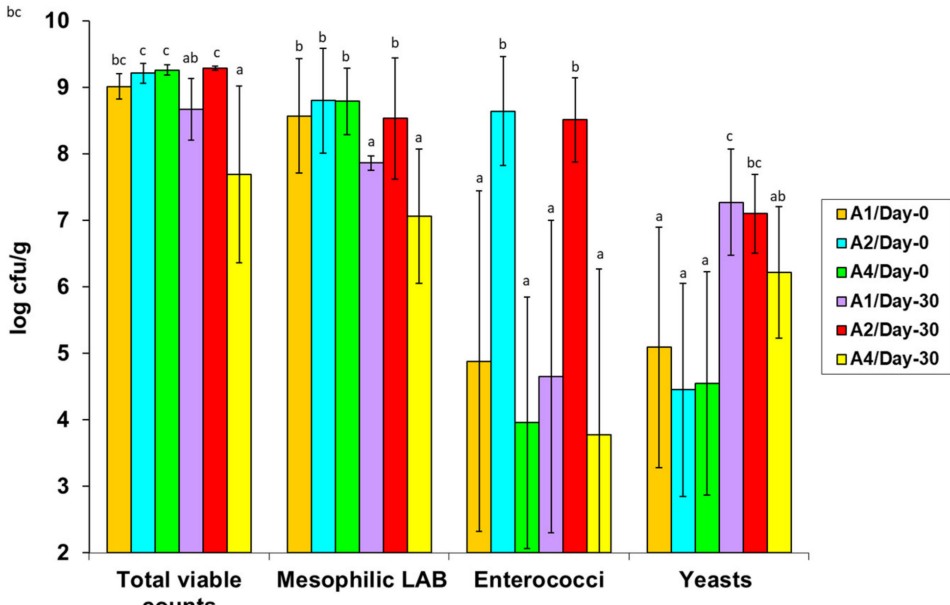

**Figure 1.** Populations (log cfu/g) of total viable bacteria, mesophilic LAB, enterococci and yeasts in fresh (day0) and cold-ripened (day30) artisan Galotyri PDO cheeses (three independent trials; *n* = 3) fermented with the A1 (ST1 + M78), A2 (ST1 + M78 + KE82) or A4 (ST1 + M78 + H25) LAB starter/adjunct strain combinations. Within each microbial group, population bars bearing different letters differ statistically (*p* < 0.05).

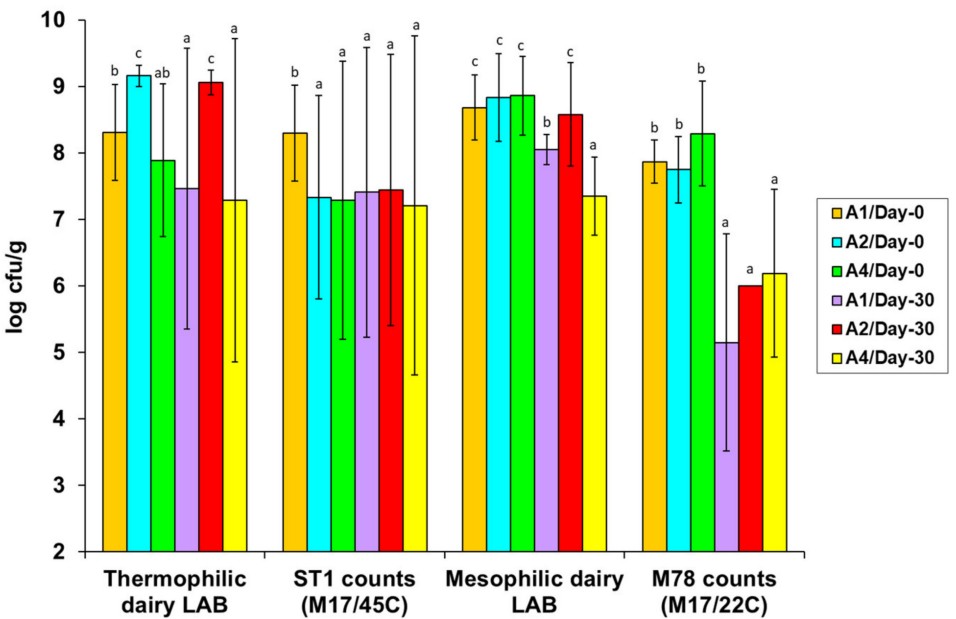

**Figure 2.** Populations (log cfu/g) of total thermophilic (counted on M17/45 °C agar) and mesophilic (counted on M17/22 °C agar) dairy LAB and the corresponding selective colony counts of the basic starter strains *Streptococcus thermophilus* ST1 and *Lactococcus lactis* subsp. *cremoris* M78 in fresh (day 0) and cold-ripened (day 30) artisan Galotyri PDO cheeses (three independent trials; *n* = 3) fermented with the A1 (ST1 + M78), A2 (ST1 + M78 + KE82) or A4 (ST1 + M78 + H25) LAB starter/adjunct strain combinations. Within each LAB group, population bars bearing different letters differ statistically (*p* < 0.05).

### 3.3.1. Effects of the Artisan Processing Technology and Hygiene on the Microbiological Quality of the Fresh Cheeses—Variations in the Populations of the Natural Starter and Adjunct Strains

As shown in Figure 1, the mean TVCs of the fresh (day 0) cheeses were slightly above 9 log cfu/g, confirming an abundant total microbial (LAB) growth at the preceding milk fermentation and acidification steps in all Galotyri cheese trials and treatments. Of note, most of the TVCs were mesophilic LAB that were capable of abundant growth at mean levels of 8.6–8.8 log cfu/g on the MRS agar with acetate (pH 5.7), including the adjunct strains KE82 and H25 in the A2 and A4 treatments, respectively (Figure 1). Specifically, uniform-pure colony lawns of the m-Ent+ *E. faecium* KE82 strain occurred on the SB/37 °C agar plates of the fresh A2 cheeses at mean populations of 8.64 ± 0.82 log cfu/g, which were numerically similar to the respective total mesophilic LAB populations (8.80 ± 0.79 log cfu/g) (Figure 1). Thus, KE82 grew abundantly in most A2 cheese milks and curds during fermentation and predominated in most fresh A2 cheeses, regardless of the presence of the basic starter (ST1+M78) strains. In contrast, in the fresh A1 and A4 cheeses, the mean populations of autochthonous enterococci were below 5 and 4 log cfu/g of cheese, respectively (Figure 1). Moreover, their growth fluctuated greatly among the fresh cheeses without KE82; it was much higher in the A1 and A4 samples in the GL1 and GL2 trials conducted in the commercial plant and the highest (6.9 log cfu/g) in the GL1/A1 cheeses. In contrast, all fresh A1 and A4 cheeses in the GL3 trial conducted in the DRD pilot plant contained <100 cfu/g of autochthonous enterococci. Regarding *L. plantarum* H25, its mean populations in the fresh A4 cheeses were 8.01 ± 0.87 log cfu/g; thus, on average, growth of H25 was ca. 10-fold lower than the total mesophilic LAB growth on the MRS/30 °C agar (Figure 1).

**Table 4.** Populations (log cfu/g) of the Gram-negative bacteria contaminating biota enumerated in three independent pilot-scale production trials of artisan Galotyri PDO cheese [1].

| Gram-Negative Bacterial Group | Cheese Trial | Artisan Galotyri Cheese Treatment | | | | | |
| | | A1 (ST1 + M78) | | A2 (ST1 + M78 + KE82) | | A4 (ST1 + M78 + H25) | |
| | | Fresh Cheese (Day 0) | Ripened Cheese (Day 30) | Fresh Cheese (Day 0) | Ripened Cheese (Day 30) | Fresh Cheese (Day 0) | Ripened Cheese (Day 30) |
|---|---|---|---|---|---|---|---|
| Pseudomonad-like bacteria (CFC agar) | GL1 | <3.00 (5.26 ± 0.11) * | <3.00 (5.72 ± 0.34) * | <3.00 (5.31 ± 0.01) * | <3.00 (5.96 ± 0.25) * | <3.00 (6.06 ± 0.07) * | <3.00 (6.12 ± 0.06) * |
| | GL2 | 7.05 ± 0.01 | <3.00 (5.54 ± 0.07) * | 5.22 ± 0.02 | <3.00 (5.78 ± 0.19) * | 3.63 ± 0.31 | <3.00 (5.13 ± 0.00) * |
| | GL3 | <2.00 | <2.00 | <2.00 | <2.00 | <2.00 | <2.00 |
| Coliform bacteria (VRB agar) | GL1 | 6.26 ± 0.09 | <1.00 | 5.40 ± 0.13 | <1.00 | 6.11 ± 0.01 | <1.00 |
| | GL2 | 6.25 ± 0.07 | 4.43 ± 0.07 | 4.83 ± 0.18 | 2.13 ± 0.39 | 3.96 ± 0.15 | <1.00 |
| | GL3 | <1.00 | <1.00 | <1.00 | <1.00 | <1.00 | <1.00 |

[1] Each value is the mean ± standard deviation of two microbial enumerations from the duplicate agar plates of each cheese sample; mean values in brackets bearing an asterisk represent the interfering antibiotic-resistant yeast populations grown on the selective *Pseudomonas* (CFC) agar. CFC, cetrimide–fucidin–cephaloridine agar; VRB, violet red bile agar (i.e., with lactose, without glucose). ST1 + M78: *S. thermophilus* ST1 + *Lc. lactis* subsp. *cremoris* M78; ST1 + M78 + KE82: *S. thermophilus* ST1 + *Lc. lactis* subsp. *cremoris* M78 + *E. faecium* KE82; ST1 + M78 + H25: *S. thermophilus* ST1 + *Lc. lactis* subsp. *cremoris* M78 + *L. plantarum* H25.

Consistent with the total mesophilic LAB on the MRS/30 °C agar (Figure 1), in all fresh cheeses, the mean populations of mesophilic dairy LAB on the M17/22 °C agar ranged at similarly high levels (8.7–8.9 log cfu/g) and showed minor differences between the A1, A2, and A4 treatments (Figure 2). Conversely, the mean populations of thermophilic dairy LAB on the M17/45 °C agar differed between the fresh cheese treatments, as they were higher in the order A2 (9.2 log cfu/g) > A1 (8.3 log cfu/g) > A4 (7.9 log cfu/g) (Figure 2). Based on the *Listeria* agar overlays conducted to clearly discriminate the inhibitory KE82 antilisterial colonies from the non-inhibitory ST1 colonies on the M17/45 °C agar plates, we confirmed that the primary starter *S. thermophilus* ST1 was overgrown by the m-Ent+ *E. faecium* KE82 adjunct strain, which prevailed in most fresh A2 cheese samples.

Contrary to our initial anticipation, ST1 grew at 9.1±0.1 log cfu/g and predominated over the NisA+ M78 co-starter strain (7.8±0.5 log cfu/g) and the two adjunct strains, KE82 (7.7 log cfu/g) and H25 (7.8 log cfu/g), in all fresh cheeses in the GL3 trial only. In fact, we decided to conduct the last GL3 trial at the DRD pilot plant after we observed the failure of ST1 to exceed 8 log cfu/g, or even grow above its inoculation level, in the fresh GL1 and GL2 cheeses produced in the commercial plant. Microbiologically, the high SD values of ST1 counts in Figure 2 reflect the aforementioned strong replicate trial effects. Amongst all fresh GL1 and GL2 cheeses, the highest ST1 growth (8.14 log cfu/g) was found in the GL1/A1 samples, whereas the lowest ST1 growth was ≤6.0 log cfu/g in the A2 and A4 cheeses in the GL2 trial. Meanwhile, in the fresh A2 cheeses in both trials, KE82 exceeded 9 log cfu/g and dominated M78 by at least 10folds, whereas H25 (8.98 log cfu/g) and M78 (9.19 log cfu/g) predominated in the fresh GL1/A4 and GL2/A4 cheeses, respectively.

However, the most unexpected finding was that neither ST1 nor M78 predominated in the A1 cheeses in the first two trials, despite their total inoculation level in the Galotyri cheese milk being as high as 6.5–7 log cfu/mL (viz. Section 2.4). Instead, both starter strains were overgrown in the fresh GL1/A1 and GL2/A1 cheeses by adventitious mesophilic LAB colony types (Figures 1 and 2), which contaminated milk after boiling, during, or after transfer in the buckets, and increased during fermentation. Diverse contaminating microbiota did not comprise autochthonous dairy LAB (Figure 2) and enterococci (Figure 1) only various Gram-negative spoilage pseudomonad-like and coliform bacteria were enumerated at growth levels as high as 7.1 and 6.3 log cfu/g, respectively, in the fresh A1 cheeses in the GL1 and GL2 trials (Table 4). The growth levels of Gram-negative bacteria contaminants in the fresh A2 and A4 cheeses were comparatively lower in both trials. Additionally, all fresh cheese samples in the GL1 trial were further contaminated with a uniform group of yeast-like colonies that could grow well (>5.2 to 6.1 log cfu/g) on the selective *Pseudomonas* agar because they were resistant to the CFC antibiotic cocktail (Table 4). Their prevalence on the CFC plates hampered the enumeration of pseudomonad-like bacteria in the fresh GL1 samples. This interfering yeast-like organism, phenotypically characterized as *Geotrichum candidum*, was retained in the ripened GL1 samples, while it later appeared at similar levels in the ripened GL2 cheeses (Table 4). Overall, fluctuating populations of diverse native spoilage yeasts occurred in the fresh cheeses of all trials at mean levels of 4.5 to 5.1 log cfu/g before storage at 4 °C (Figure 1). The CFC-resistant *G. candidum* group (Table 4) was a subdominant fraction of the total yeast populations in the fresh A1 (7.1 log cfu/g), A2 (6.2 log cfu/g), and A4 (6.4 log cfu/g) cheese samples in the GL1 trial. Conversely, due to the hygienic precautions during Galotyri processing in the DRD pilot plant, all fresh GL3 cheese samples were free of spoilage Gram-negative bacteria contaminants (Table 4), but they contained considerable (4.2–4.7 log cfu/g) CFC-sensitive yeast contamination levels before cold ripening.

3.3.2. Effects of Cold Ripening on the Microbiological Characteristics of the Ripened Cheeses—Variations in the Surviving Populations of the Natural Starter and Adjunct Strains

Significant changes in microbial populations occurred after aerobic (cold ripening) storage of the fresh RTE cheeses at 4 °C for 30 days. The most pronounced change was the major

growth increases of native yeasts in all ripened cheeses at mean levels of 6.2–7.3 log cfu/g; of note, the final yeast populations were constantly lower in the A4 compared to the A1 and A2 ripened (day 30) cheeses (Figure 1). Contrary to the yeasts, no LAB group, including the inoculated LAB strains, promoted growth during cold ripening; however, all LAB survived, either with minor or major declines. Therefore, on average, TVCs and the predominant mesophilic LAB populations decreased significantly from day 0 to day 30 in all cheeses, except for the A2 cheeses, because the predominant m-Ent+ KE82 strain survived with minor (<0.5 log) death on the selective SB/37 °C agar and the other four LAB enumeration agar media, i.e., MPCA/37 °C, MRS/30 °C, M17/22 °C, and M17/45 °C (Figures 1 and 2). Of special note, ST1 also survived without death in all ripened GL3 cheese samples, i.e., the only fresh trial in which ST1 grew predominantly (>9 log cfu/g). In contrast, ST1 declined to or was below the 6-log inoculation level in the ripened GL1 and GL2 cheese samples. The above major variations are reflected by the high SD values of the ST1 counts in Figure 2. M78 also underwent major (>2.2–4 log units) declines in all ripened cheeses (Figure 2), particularly in those of the best-monitored GL3 trial, a result suggesting that the NisA+ *Lc. lactis* strain might have autolyzed after entering the stationary phase at the low pH (<5.0) of the Galotyri Acid-Curd cheeses during cold ripening. With regard to the H25 adjunct strain, its populations declined from 10 folds to 100 folds in the ripened A4 cheeses in the first two trials but not in the ripened GL3/A4 cheeses, where they remained constant at 7.8 log cfu/g after 30 days of cold ripening storage at 4 °C.

Gram-negative bacteria remained below detection in all cold-ripened GL3 cheeses, while coliform bacteria died off in all GL1 cheeses but not in all GL2 cheeses (Table 4), which had pH≥4.6 across processing. Specifically, on day 30, coliform survival was the greatest (ca. 4.5 log cfu/g) in the GL2/A2 cheeses (pH 4.8) versus their inactivation (<10 cfu/g) in the GL2/A4 cheeses (pH 4.6). Likewise, the high initial (day 0) populations of pseudomonad-like bacteria declined in all ripened GL2 cheeses, but their actual survival, if any survived, could not be estimated because the CFC agar plate lawns were full of the interfering CFC-resistant *G. candidum* colonies (Table 4). Although *G. candidum* and other yeast contaminant species might have consumed some lactate formed by LAB in the fresh GL1 (pH 4.4–4.5) and mainly GL2 (pH 4.6–5.0) cheeses, the pH was not reversed in any cheese trial or treatment (Table 3) due to major lactate assimilation by non-LAB or yeasts from day 0 to 30 [48,53]; this was an important positive finding regarding BA formation and the total microbial quality and safety of the cold-ripened Galotyri cheeses.

*3.4. Major Variations in the Biogenic Amine (BA) Amounts of the Artisan Galotyri Cheese Samples Associated with the Microbial Quality of the Fresh Cheeses and Cold Ripening Effects*

The results on the detection and quantification of the seven BAs in the fresh (day 0) or ripened (day 30) artisan Galotyri cheese samples are presented separately for each of the three independent pilot plant trials in Table 5 due to the strong trial-dependent replicate effects critically interpreted in Sections 3.2 and 3.3 above.

Indeed, in most cases, the net differences in BA concentrations between the GL1, GL2, and GL3 trials for a given cheese sampling time (day 0 or 30) or LAB (A1, A2, or A4) treatment were so high that the statistical analyses of the calculated BA mean values were invaluable (i.e., the SD values were too high). Nevertheless, as shown in Table 5, the separate evaluation of the BA amounts quantified within the cheese samples of each trial leads to important findings regarding certain differences between treatments. The most profound findings for all cheese trials, with exceptions highlighted, are as follows.

The fresh A2 cheeses contained very low to undetectable amounts of BAs, particularly tyramine (Table 5), despite the high predominance in most A2 samples of the m-Ent+ *E. faecium* KE82 adjunct strain (Figure 1) previously shown to form abundant tyramine in cultured BA broth in vitro (Table 1). Relative to the aims of this study, the above constant result was the most positive finding, contrary to the literature and our anticipation.

**Table 5.** Biogenic amine concentrations (mg/kg) of three independent artisan Galotyri PDO cheese trials produced with three Greek indigenous (A1, A2, and A4) starter/adjunct LAB strain combinations.

| Biogenic Amine | Cheese Trial | Artisan Galotyri Cheese Treatment | | | | | |
|---|---|---|---|---|---|---|---|
| | | A1 (ST1 + M78) | | A2 (ST1 + M78 + KE82) | | A4 (ST1 + M78 + H25) | |
| | | Fresh Cheese | Ripened Cheese | Fresh Cheese | Ripened Cheese | Fresh Cheese | Ripened Cheese |
| Cadaverine | GL1 | 18.1 | NT | 4.7 | NT | 21.9 | NT |
| | GL2 | 158.6 | 3521.1 | <0.5 | 5664.8 | 815.1 | 3350.4 |
| | GL3 | 242.6 | 5.8 | 4.0 | 4.1 | 4.0 | 4.4 |
| Histamine | GL1 | 245.9 | NT | <0.5 | NT | <0.5 | NT |
| | GL2 | 503.5 | 87.5 | <0.5 | 373.1 | 179.8 | 145.9 |
| | GL3 | 130.2 | 15.3 | 14.9 | 16.6 | 14.0 | 16.4 |
| Putrescine | GL1 | <0.5 | NT | <0.5 | NT | <0.5 | NT |
| | GL2 | <0.5 | 897.3 | <0.5 | 1396.7 | 373.3 | 278.4 |
| | GL3 | 570.9 | <0.5 | <0.5 | <0.5 | <0.5 | <0.5 |
| Spermidine | GL1 | <0.5 | NT | <0.5 | NT | <0.5 | NT |
| | GL2 | 74.4 | <0.5 | <0.5 | 34.3 | 21.6 | 13.1 |
| | GL3 | 60.4 | 9.4 | 9.2 | 9.3 | 9.2 | 9.2 |
| Tryptamine | GL1 | 10.9 | NT | 1.1 | NT | 0.8 | NT |
| | GL2 | <0.5 | 401.3 | <0.5 | <0.5 | 99.5 | <0.5 |
| | GL3 | 79.9 | 6.0 | 5.0 | 7.0 | 4.6 | 6.9 |
| Tyramine | GL1 | 26.3 | NT | 21.8 | NT | 4.4 | NT |
| | GL2 | 591.0 | 90.0 | 3.7 | 134.1 | 165.2 | 61.8 |
| | GL3 | 103.3 | 3.9 | 7.0 | 3.9 | 4.6 | 4.0 |
| 2-Phenylethyl Amine | GL1 | <0.5 | NT | <0.5 | NT | <0.5 | NT |
| | GL2 | 251.6 | 140.1 | <0.5 | 125.9 | 1989.5 | 34.1 |
| | GL3 | 65.5 | 1.3 | 1.5 | 2.7 | 1.4 | <0.5 |

NT, not tested. ST1 + M78: *S. thermophilus* ST1 + *Lc. lactis* subsp. *cremoris* M78; ST1 + M78 + KE82: *S. thermophilus* ST1 + *Lc. lactis* subsp. cremoris M78 + *E. faecium* KE82; ST1 + M78 + H25: *S. thermophilus* ST1 + *Lc. lactis* subsp. *cremoris* M78 + *L. plantarum* H25.

Thus, unexpectedly, all fresh A2 (ST1 + M78 + KE82) cheeses contained either similar or, in most cases, much lower BA amounts than the A4 (ST1 + M78 + H25) cheeses; especially, the fresh GL2/A4 cheeses contained much higher amounts (mg/kg) of cadaverine (815.1), putrescine (373.3), histamine (179.8), tyramine (165.2), tryptamine (99.5), spermidine (21.6), and 2-phenylethylamine than the fresh GL2/A2 cheeses, which contained only 3.7 mg/kg of tyramine and traces (<0.5 mg/kg) of the other six BAs (Table 5).

Also, unexpectedly, the fresh A1 (ST1 + M78) typical (control) Galotyri cheeses contained more BAs in higher amounts than those quantified in the fresh A4 (in many cases) and A2 (always) new specialty cheeses. This difference was most prominent in the microbiologically superior GL3 cheeses; only the fresh A1/GL3 cheeses contained significant amounts of all seven BAs tested, mainly putrescine (570.9 mg/kg), while histamine (130.2 mg/kg) and tyramine (103.3 mg/kg) were present at intermediate amounts (Table 5).

Clearly, the worst trial with regard to the total BA content of the fresh cheeses was GL2, in correlation with its inferior microbial quality and the highest levels of Gram-negative bacteria in the fresh GL2/A1 cheeses (Table 4), which also contained the highest amounts of histamine (503.5 mg/kg) and tyramine (591.0 mg/kg) among all cheese samples (Table 5). Moreover, in most GL2 cheese samples, the BA concentrations increased greatly from the fresh to the ripened cheese after 30 days of storage at 4 °C. The greatest increases after cold ripening were detected for cadaverine and putrescine in the GL2/A1 and mainly the GL2/A2 cheeses and for tryptamine in the GL2/A1 cheeses only. Also, the ripened GL2/A2 cheeses displayed increases in the amounts of histamine (373.1 mg/kg), tyramine (134.1 mg/kg), and 2-phenylethylamine (125.9 mg/kg) compared to the very low to trace

amounts they contained before ripening (Table 5); the association of this result with the great survival, without death, of the KE82 adjunct strain during cold ripening is uncertain.

Otherwise, except for the few cheese sample occasions indicated above, cold ripening appeared to have minor adverse effects on increasing the BA content of the cheeses. Instead, compared to the BA concentrations quantified in the fresh cheeses on day 0, major to moderate decreases in the amounts of all seven BAs occurred in the ripened GL3/A1 cheeses, as well as in the ripened GL2/A4 cheeses (Table 5).

## 4. Discussion

The great fluctuations in the BA amounts noted between the three independent pilot plant Galotyri cheese trials, as well as between the three starter/adjunct LAB culture treatments within each trial, are in agreement with the relevant literature. Generally, BA production is a complex, multi-factor-dependent biochemical process that is difficult to predict and control in fermented (dairy) foods [2,3,7,67]. Particularly in cheese, each individual BA may fluctuate from undetectable levels of a few mg/kg to several g/kg [1,4,6,68,69]. Major fluctuations often exist between similar cheeses from different plants [31], different productions of the same cheese in one plant, or even different locations in one cheese product or unit [70,71]. Similarly, in this study, major BA-dependent fluctuations were noted between the fresh GL1 and GL2 cheeses, despite all being processed in the same commercial plant environment under similar artisan cheese manufacturing and hygienic conditions; the milking season was the only prominent difference, as trial GL1 was processed in June and GL2 in December. Unexpectedly, major BA-dependent fluctuations occurred between the A1, A2, and A4 treatments of the fresh GL3 cheeses. Moreover, despite the fact that the GL3 trial was processed in the DRD pilot plant under strict hygienic conditions, certain BAs, such as putrescine (570.9 mg/kg), cadaverine (242.6 mg/kg), and tryptamine (79.9 mg/kg), were at higher initial concentrations in the fresh A1/GL3 cheeses compared to the respective A1 cheeses in the GL1 and GL2 trials. This finding was difficult to explain because none of the GL3 cheeses contained detectable levels of Gram-negative bacteria (Table 4). Also, neither *Lc. lactis* subsp. *cremoris* M78 produced putrescine in pure culture in control BA broth (i.e., without ornithine but containing tryptone, yeast extract, and meat extract [58]) (Table 1) nor in coculture with *S. thermophilus* ST1 in SRM (viz. Section 3.1), unlike several dairy *Lc. lactis* strains tested by others [1,11,37,72]. Thus, the high initial contents of the above three BAs in the fresh A1/GL3 cheeses might be due to the natural yeast contamination and growth [22,73,74] that occurred during the draining of the fresh cheese curds in the DRD pilot plant. Conversely, the too-high increases in putrescine (>800 mg/kg) and mainly cadaverine (>3000 mg/kg) that occurred in most GL2 cheeses during cold ripening were attributed to the major growth of Gram-negative bacteria (>6.0 to 7.1 log cfu/g) in the freshly fermented cheeses (Table 4). Probably, the decarboxylases of Gram-negative bacteria secreted in the soft cheese mass during fermentation remained active during storage at 4 °C to act together with decarboxylases potentially formed by the yeast populations [22,73] grown in the cold ripening GL2 cheeses. Conversely, the positive effects of strict hygiene on the microbial quality of the fresh GL3 cheeses were reflected by the total suppression of Gram-negative bacteria (Table 4) and, consequently, by the major reductions of all BA contents in the ripened A1/GL3 cheeses and by the lack of BA accumulation in the ripened A2/GL3 and A4/GL3 cheeses, respectively (Table 5).

Technologically, the BA accumulation potential is higher in ripened cheeses made of raw milk compared to similar products made from pasteurized milk [75], as well as in long-ripened hard cheeses and mold-ripened soft cheeses compared to other cheese types [1,4]. Because BAs are formed by the enzymatic decarboxylation of precursor amino acids or the amination or transamination of aldehydes and ketones [4,5], their formation depends strongly on cheese proteolysis, which usually takes place during ripening [75–79]. The longer the ripening period or the higher the implication of molds in the ripening process, the greater the cheese proteolysis index and, thereby, the BA formation potential. Also, the higher the population of decarboxylating microorganisms in cheese, the higher the BA

formation and accumulation potential, reflecting why raw milk cheeses subjected to natural LAB fermentations are at higher risk [1,4,75]. As a general rule, the presence of dangerous BA amounts in foods is associated with decarboxylating microbial growth levels above 7 log cfu/g [3], which can be reached in fermented cheese quite easily [75]. In this study, the ewes' milk used was boiled and thus it was free (<10 cfu/mL) of bacterial contaminants after boiling. Moreover, BAs were not detected in raw milk after sterilization or boiling, confirming previous reports by others on the low to undetectable BA levels in milk [5,80]. However, the use of improperly disinfected buckets, particularly plastic ones, to ferment the boiled milk slowly at ambient temperatures and of cheesecloth bags to drain the fresh Galotyri curds were two primary contamination sources with a spontaneous LAB biota intermixed with diverse spoilage Gram-negative bacteria and yeasts, particularly from the commercial plant environment [48]. Therefore, despite inoculation with the selected ST1, M78, KE82, and H25 strains, those indigenous NSLABs, along with fast-growing Gram-negative spoilers and later cold-tolerant yeasts, predominated in the fresh and ripened GL2 cheeses, which thus accumulated BAs, cadaverine, and putrescine, followed by histamine, tyramine, 2-phenyethylamine, and tryptamine (Table 5). Similar results have been reported in the literature regarding the association of high BA amounts in cheese with high levels of Gram-negative bacteria, mainly *Enterobacteriaceae* [14–16]. Overall, unless the milk fermentation process deviates, as in the GL2 trial, fresh cheeses normally have lower BA amounts than ripened cheeses, either due to reduced growth of BA-forming microbiota [81] or lower availability of precursor amino acids because of less proteolysis [8]. Carefully selected starter cultures that include no decarboxylating strains not only prevent LAB growth deviations and BA production during (fresh) cheese fermentations but may also degrade the BAs formed by indigenous NSLAB strains during ripening to an extent, depending on their BA detoxification capabilities, which probably happened in the A1/GL3 cheeses (Table 5).

Apart from milk heating (pasteurization) and starter cultures, additional processing factors, such as temperature, pH, NaCl concentration, and time and temperature of ripening and/or preservation, are critical for the BA accumulation potential in cheese [3,4]. In general, elevated temperatures during processing, ripening, and/or storage enhance microbial growth and thus BA formation. In this study, the temperature was elevated only during the first 3 to 6 fermentation hours of the pre-boiled, LAB-inoculated Galotyri cheese milk, after which the temperature fell to ca. 20–22 °C, and then to 12 °C during curd draining, and finally to 4 °C during storage of the fresh RTE cheeses [48]. Therefore, accelerated temperature-dependent BA production could only be associated with the fast and abundant growth of Gram-negative bacteria contaminants during the fermentation in the GL2 cheese milk with specific decarboxylating activities of the psychrotrophic pseudomonad-like bacteria and enterobacteria during all subsequent cheese processing (i.e., drying, salting, and packaging) and cold ripening steps.

Research data on the effects of pH on BA production and potential accumulation in cheese are rather controversial. An acidic pH inhibits the growth of several decarboxylating groups of spoilage Gram-negative bacteria and Gram-positive non-LAB, but, on the other hand, BA formation is triggered at the bacterial cellular level as an 'internal neutralization' response to counteract the external acid stress caused by organic acid formation in food (cheese) fermentations [82]. Of note, most amino acid decarboxylases have optimal activity at pH around 5.0 [1,4]; hence, BA formation is expected to be more pronounced in fermented or acidified cheese (pH 4.5–5.5) than in milk or fresh non-fermented (whey) cheese environments (pH 6.0–6.7). On this basis, the acid pH range (i.e., 4.0–5.0) of the artisan Galotyri PDO cheese fermentations in the present and previous studies and, generally, the low pH of most traditional (Greek) Acid-Curd cheeses may be favorable for optimal BA production. Therefore, the most effective preventive measures to avoid the BA risk in fermented (low pH) cheeses are (i) the GMP and GHP during processing; (ii) the addition of well-defined starter cultures in pasteurized milk; (iii) the lowering of milk fermentation

and curd drying temperatures; and (iv) the storage of fresh RTE cheeses under refrigeration until consumption [8,31,74,83].

Regarding salt, high NaCl concentrations (≥5%) inhibit BA production, most likely due to the inhibitory effect of the high salt contents on the growth rate of BA-producing bacteria [24,77,84]. To date, however, most fermented cheeses have remarkably lower salt contents (usually 1–3%) due to consumer health and diet considerations, which, in turn, may enhance BA production in cases where the microbial quality of the milk is inferior and/or the cheese processing, ripening, or storage conditions favor decarboxylation. In this study, favorable conditions for decarboxylation were established in the fresh cheeses in the GL2 trial only, most of which accumulated high BA amounts, mainly cadaverine and putrescine, during ripening despite their salt content (2.3–3.2%) being the highest among all cheese trials. The high variability in the salt content between trials (Table 3) was due to the fact that the artisan drying and salting operations of the freshly fermented Galoryri cheese curds in the commercial plant were monitored poorly [48,53].

Based on the variability of the results shown in Table 5, the safety of artisan Galotyri cheese products with regard to their potential BA toxicological effects cannot be adequately addressed [85]. As reported in the Introduction, maximum legal BA levels have not yet been established for cheese and other dairy products [6,85] and, overall, for any BA in any food, except for histamine in fishery products [12,13]. One main reason for the lack of specific legislation for BAs in foods is that BAs are biosynthesized in mammalian (human) cells in order to act as precursors for the synthesis of hormones or play an important role in other biological functions, while the toxicity levels also vary greatly from person to person [1,85]. Nevertheless, considering their total BA content (Table 5), the artisan Galotyri cheeses in the GL2 trial should be regarded as dangerous, irrespective of ripening, whereas the GL1 and GL3 cheeses would be safe, with only a few of them having the potential to become toxic to sensitive people. Specifically, according to the EFSA [13], histamine concentrations over 500 mg/kg are harmful to health, and more than 1000 mg/kg can be lethal for sensitive people. However, concentrations of histamine over 500 mg/kg have recently been detected in commercial Spanish cheeses [31], whereas the toxicological thresholds for histamine in dairy foods to be considered safe for all people should be 10folds lower, 50–100 mg/kg. Hence, based on their histamine content, the fresh A1/GL2 cheeses could be harmful, followed by the ripened A2/GL2 cheeses.

The maximum allowable quantities of tyramine in foods in Austria were estimated by Paulsen et al. [86], who recommended a NOAEL (i.e., No Observed Adverse Effect Level) of 200 mg of tyramine per meal for people who are not susceptible to tyramine. Cheese, in particular, may have maximum allowable tyramine levels of 1000 mg/kg [86], while the toxicological threshold reported by the EFSA for tyramine is 600 mg/kg [13]. According to other reports, tyramine consumption normally ranges from 100 to 800 mg/kg [1], with levels over 1080 mg/kg being hazardous. However, people who receive monoamine oxidase inhibitor medication are more susceptible to tyramine consumption; a diet containing 60 mg/kg may result in a mild crisis, whereas a diet containing 100–250 mg/kg is linked to a severe crisis [86]. Therefore, based on their tyramine content, all Galotyri cheese products in this study could be regarded as safe; however, the fresh A1/GL2 cheeses might cause severe crises for susceptible individuals.

Amines that may exacerbate the negative effects of histamine and tyramine include putrescine and cadaverine, which are aliphatic diamines that can build up in significant quantities (>1.5 to >3 g/kg) in many foods, particularly cheese [13]. Therefore, putrescine and cadaverine are among the most prevalent BAs present in cheeses, along with tyramine and histamine [1,67,87]. Additionally, both have been reported as potentiators of the toxic effects of other amines due to the inhibition of detoxifying enzymes [77]. Published data on the toxicity of putrescine and cadaverine are scarce; no human dose–response data are available, and only one animal study has been published in which a NOAEL of 2000 ppm (180 mg/kg body weight/day) was established in Wistar rats [88].

Consequently, no legal limit has been established for putrescine in any food, even though this BA may accumulate in cheese at concentrations as high as 1560 mg/kg [13], a level much higher than the lowest concentration of putrescine found to be cytotoxic (NOAEL = 10 mM, equivalent to 881.5 mg/kg) by Del Rio et al. [89]. No legal limit has been established for cadaverine in any food either, even though this BA may also accumulate at very high concentrations in cheese (up to 3170 mg/kg) [13]; thus, levels much higher than the lowest concentration were found to be cytotoxic (NOAEL = 5 mM, equivalent to 510.89 mg/kg) [89]. Hence, the above concentrations should be considered hazardous to human health. Based on toxicological information and consumption habits in Austria, Rauscher-Gaberniget et al. [90] suggested lower to similar tolerable levels for putrescine and cadaverine in various foods than the aforementioned concentrations. Specifically, for putrescine in cheese, the proposed maximum tolerable level was 180 mg/kg, whereas that for cadaverine was 540 mg/kg. Altogether, the above data support the idea that nearly all cheeses from the 'deviated' GL2 trial could be toxic due to the accumulation of high putrescine and cadaverine amounts that were too high, particularly after one month of cold ripening storage.

Additional studies showed that tyramine and histamine have synergistic cytotoxicity toward intestinal cell cultures and that histamine, below the legal limit, can raise the cytotoxicity of tyramine at a concentration frequently present in foods [91]. Because, as mentioned, BA poisoning varies based on each person's susceptibility and whether other amines are present, Ladero et al. [92] proposed a maximum total BA level of 750–900 mg/kg, which, in this study, was much exceeded in nearly all unsafe GL2 cheeses and, surprisingly, only in the fresh A1/GL3 cheeses in the other trials. Regarding polyamines (PAs), such as spermine and spermidine, no limits are proposed either, although they are also cytotoxic to intestinal cell cultures [93]. However, PAs are naturally present in foods, including milk [1,80], at low levels that are not likely to be harmful to healthy people [93]; additionally, they are not dietary BAs formed microbiologically [1]. Spermidine did not seem to be important in the fresh or ripened Galotyri cheeses either (Table 5).

The most important finding regarding artisan Galotyri cheese safety was the lack of correlation of the A2 (ST1 + M78 + KE82) starter/adjunct culture with increased tyramine production in any pilot plant trial, including the unsafe cheeses in the GL2 trial (Table 5). Despite the fact that m-Ent+ *E. faecium* KE82 produced abundant tyramine in BA broth with 1% tyrosine in vitro (Table 1), this undesirable trait was not confirmed in the model A2/SRM cocultures. Also, no major production or accumulation of tyramine was detected in any of the A2 cheese samples, despite KE82 growing well (>7.5 log cfu/g) in all Galotyri trials and particularly predominating (>8.5 log cfu/g) the A2/GL1 and A2/GL2 cheeses. Although additional studies of traditional Greek cheeses ripened at higher temperatures for longer times are required to assure low in situ tyramine formation by KE82, the use of this strain in artisan Galotyri cheese appears safe with regard to BA formation, provided that GMP and GHP are followed. Under GHP in cheese plants, like those applied in the DRD pilot plant (trial GL3), the use of the m-Ent+ *E. faecium* KE82 as an antilisterial adjunct [53,55] may deliver further biotechnological benefits relating to its moderate to strong esterase–lipase and aminopeptidase activities [54] and potential probiotic properties [94,95]. Functional cheese studies incorporating advanced biochemical methods and qrt-PCR of the in situ decarboxylating gene expression and production are required to demonstrate the above benefits, without overlooking that all *Enterococcus* species neither have a GRAS status nor are included in the Qualified Presumption of Safety (QPS) list of food microorganisms [96]. Therefore, the use of m-Ent+ Greek cheese isolates of the *E. faecium/durans* group in the dairy industry is currently prohibited. Despite none of them being β-hemolytic or vancomycin-resistant or harboring any of the common enterococcal virulence genes, such as *agg*, *ace*, *espA*, *IS16*, *hyl*, or *gelE* [54], their application in commercial Galotyri productions should be approached with caution because even safe *Enterococcus* strains may transfer virulence factors and antibiotic-resistance genes (for instance, KE82 is resistant to erythromycin and penicillin in vitro [54]) and can have adverse effects on human health [95].

Unlike all *Enterococcus* spp., the application of *L. plantarum* in food production has become widespread, owing to its GRAS status and the superior biotechnological, antipathogenic, antifungal, and probiotic properties of many selected strains [97]. Although *L. plantarum* typically is an NSLAB in dairy fermentations, since 2000, numerous strains have been applied as CSC constituents or (probiotic) adjunct cultures in various cheese types [97,98], including functional Greek Feta [99] and Galotyri PDO [46–48,52] cheeses. Regarding BA formation, *L. plantarum* is among the LAB species tabulated as histamine, tyramine, and putrescine producers in the recent relevant review by Barbieri et al. [11]. However, most BA-forming *L. plantarum* strains are linked to fermented sausages [11], whereas the implication of this species as a primary BA producer in cheese and other fermented milk products is limited compared to *L. curvatus*, *L. casei*, or gas-forming lactobacilli, such as *L. buchneri*, *L. parabuchneri*, and *L. brevis*, which also are NSLABs in dairy foods [4,11]. Only one out of the sixteen *L. plantarum* strains originally assayed using the improved decarboxylase screening broth was found to be positive for at least one BA versus 12/15 *L. curvatus*, and 3/4 *L. brevis* and 1/1 *L. buchneri* strains were positive [58]. Conversely, Espinosa-Pesqueira et al. [18] retrieved three and four *L. plantarum* strains from Spanish artisan goat's and ewe's raw milk cheeses that produced 307.6–528.5 mg/L and 45.1–353.3 mg/L tyramine, respectively, in decarboxylase (BA) broths in vitro, whereas all produced low (<27 mg/L) to undetectable amounts of histamine, putrescine, and cadaverine. Moreover, in situ tyramine production by an *L. plantarum* strain in yogurt was detected by Yilmaz and Görkmen, who further noted that the accumulation of tyramine was enhanced by possible synergistic interactions between the yogurt starter *L. delbrueckii* subsp. *bulgaricus* and tyramine-forming LAB [100]. Also, in this study, there was evidence that the adjunct strain *L. plantarum* H25 might be responsible for a restricted mean (48.6 mg/kg) tyramine production in coculture with the basic starter ST1+M78 strains in SRM (viz. Section 3.1), corroborating its ability to produce 35.1–187.1 mg/L of tyramine in pure culture in BA broth, irrespective of 1% tyrosine addition (Table 1). Therefore, the above low to moderate tyramine productions may associate with strong inherent proteolysis by the H25 strain in the presence of tryptone in BA broth and of milk proteins in SRM, concurrently with its very strong-positive leucine, valine, and cysteine arylamidase and other enzymatic activity reactions determined by the API ZYM method [54]. Although the major (>7 log cfu/g) growth of *L. plantarum* H25 in the fresh A4/GL1 and mainly in the fresh and ripened A4/GL3 cheeses was not correlated with increases in tyramine, further studies are needed to ensure that neither H25 nor other indigenous *L. plantarum* strains selected as potential probiotics during the BIOTRUST project are in situ tyramine or other BA producers before their commercial application in local traditional dairy plants in Epirus or Thessaly.

Surprisingly, BA studies on traditional Greek cheeses, either as RTE retail products or as a function of the cheese ripening process, are scarce [77,101]. In 2000, Valsamaki et al. [77] published the first report on BA production in Feta cheese, noting that the total BA content in mature 60-day-old cheese was only 330 mg/kg and increased to 620 mg/kg after 120 days of storage. The main BAs were tyramine and putrescine (69.7% and 71.2% at 60 and 120 days, respectively; ca. 200 mg/kg each), followed by histamine (90 mg/kg), while tryptamine and phenylethylamine concentrations were all very low with regard to ripening [77]. Interestingly, BA production was biphasic; it was major from 1 to 15 days and from 60 to 120 days of ripening. The relatively low total BA levels were attributed to the low pH and high salt content of Feta cheese, which appeared to create unfavorable conditions for amino acid decarboxylation [77]. However, no microbiological data were provided to correlate the BA levels with the number and type of LAB prevailing in Feta cheese and with, most likely, a low presence of undesirable decarboxylating Gram-negative bacteria and yeasts during processing or ripening. No microbiological data were reported by Zotou and Notou [101] either, who measured BA mixtures in Greek Feta and Kasseri cheeses (i.e., three commercial brands each) as an application to verify the utility of an advanced derivatization and fluorescence HPLC determination method. In Feta cheese, tyramine (56–130 mg/kg)

was the most abundant BA, followed by histamine (14.9–78.6 mg/kg), while in Kasseri cheese, histamine, and particularly tyramine, were at lower concentrations (<40 mg/kg). Putrescine, cadaverine, tryptamine, and 2-phenylethylamine were not detected in any of the cheese samples [101], suggesting that industrial Feta and Kasseri brand cheeses with low, if any, *Enterobacteriaceae* or yeast contamination were analyzed. In contrast, artisan Feta and other Greek cheeses may often harbor quite high (>6 to 7 log cfu/g) populations of Gram-negative bacteria, mainly enterobacteria, when still they are freshly fermented [44,102], owing to poor sanitary conditions during cheese making; this defect may increase BA, mainly cadaverine, formation [1,4,7,16], as noted in the present GL2 Galotyri cheeses (Tables 4 and 5). High (>6 to 7 log cfu/g) yeast populations may also develop in ripened Feta and other white-brined or Acid-Curd artisan Greek cheeses [52,74], as noted in all present Galotyri cheeses, too (Figure 1).

**5. Conclusions**

This study is the first research report which associate BA formation in an artisanal Greek PDO cheese production, i.e., the oldest Acid-Curd cheese Galotyri, with the use of native starter/adjunct LAB strain cultures and the microbiology of the resultant cheeses after processing (fresh RTE cheese) and cold ripening for 30 days. The results showed that one of the fresh cheese batches accumulated high amounts of total BAs, mainly cadaverine and putrescine, which increased further in the cold-ripened cheeses. Hazardous BA levels were associated with an outgrowth (>7 log cfu/g) of post-thermal Gram-negative bacteria contaminants during the fermentation of that particular fresh cheese. Conversely, none of the starter or adjunct LAB strains could be correlated with a specific BA increase, despite the fact that *E. faecium* KE82, included in one of the LAB combinations, is a strong tyramine producer in vitro. In conclusion, while the high occurrence of Gram-negative decarboxylating bacteria compromised cheese safety, the adoption of strict hygienic measures during artisan Galotyri production reduced BA formation in hygienically cold-ripened cheeses. Overall, research on BA formation during the fermentation and ripening of traditional Greek cheeses should be intensified, as was performed in other Mediterranean countries with a long tradition of cheese production. Advanced research studies are required to correlate BA formation with specific bacteria and yeast species or strains and establish control measures toward BA accumulation in traditional Greek cheeses, with an emphasis on the application of novel functional starter cultures containing BA-degrading LAB or yeast strains.

**Author Contributions:** Conceptualization, J.S.; methodology, J.S., C.T., and L.B.; formal analysis, J.S., C.T., L.B., and A.K.; resources, J.S.; data curation, J.S. and L.B.; writing—original draft preparation, J.S. and L.B.; writing—review and editing, J.S. and L.B.; project administration, J.S. All authors have read and agreed to the published version of the manuscript.

**Funding:** This research was funded by the European Union and Greek national funds through the EPAnEK 2014–2020 Operational Program Competitiveness, Entrepreneurship, and Innovation under RESEARCH-CREATE-INNOVATE (project T1EDK-00968; project acronym BIO TRUST).

**Institutional Review Board Statement:** Not applicable.

**Informed Consent Statement:** Not applicable.

**Data Availability Statement:** Data are contained within the article.

**Conflicts of Interest:** The authors declare no conflicts of interest.

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
