# Peer review of "Biogenic Amine Formation in Artisan Galotyri PDO Acid-Curd Cheeses Fermented with Greek Indigenous Starter and Adjunct Lactic Acid Bacteria Strain Combinations: Effects of Cold (4 °C) Ripening and Biotic Factors Compromising Cheese Safety"

_2673-8007, doi:10.3390/applmicrobiol4010038_

Round 1

Reviewer 1 Report

Comments and Suggestions for Authors

Line 40 what is the primary factor of BA accumulation in cheese?

Line 109 what is Direct-Vat-Set?

Remove “4o”C in the title.

In materials and method, please provide flow chart of cheese production using those combinations.

Line 215, why the incubation period was extended up to 66 h?

What is the reason of using strains with specific function such as nicin-producing strain or strain carry m-Ent?

What is the acceptable dose of BA that may occur in cheese? Since in your results (Table 5) show that most of the results have produced BA.

Why the protein content between fresh and ripened cheese not to significant?

During cold ripening, did the authors also measure the moisture (RH)?.

Line 678, “in agreement with the relevant literature”, please mention the literature and what are their statement/findings.

Line 694, since the result of GL3 is hard to answer, what is your strategy to cope this issue?

Prior using the strains, did the author have reveal the whole genome sequences of the strains to gain more insight/potential of the strains to produce BA?

Comments on the Quality of English Language

Minor English is required

Author Response

Dear Reviewer 1,

Please refer to the attached PDF file with our responses to your comments. Thank you.

John Samelis

Reviewer 2 Report

Comments and Suggestions for Authors

The manuscript titled “Biogenic Amine Formation in Artisan Galotyri PDO Acid-Curd Cheeses Fermented with Greek Indigenous Starter and Adjunct Lactic Acid Bacteria Strain Combinations: Effects of Cold (4℃) Ripening and Biotic Factors Compromising Cheese Safety” evaluated the effects of Cold (4℃) Ripening and Biotic Factors Compromising Cheese Safety. However, specific points have to be addressed as follows for authors’ consideration.

1/ The section Abstract of this manuscript needs to be modified. An abstract is a summary presentation of the innovative findings of your work, as simple as possible, rather than a brief description of methods and results.

2/ Please control the number of keywords.

3/ Line 154, four combinations (A1, A2, A3, A4), it is better to use the table to show these four combinations than to use words.

4/ In the section 2.7. Determination of Biogenic Amines in SRM and Galotyri Cheese Samples, no formula regarding the amount of the analytes is presented here.

5/The section Conclusion should be supplemented for this manuscript.

6/Table legends are very long.

Author Response

Dear Reviewer 2,

Please refer to the attached PDF file with our responses to yuor comments. Thank you.

John Samelis
